# Simultaneous Quantification of Five Sesquiterpene Components after Ultrasound Extraction in *Artemisia annua* L. by an Accurate and Rapid UPLC–PDA Assay [note 1]

**DOI:** 10.3390/molecules24081530

**Published:** 2019-04-18

**Authors:** Jiaqi Ruan, Zhengyue Liu, Feng Qiu, Henan Shi, Manyuan Wang

**Affiliations:** School of Traditional Chinese Medicine, Capital Medical University, No.10, Xitoutiao, You’anmenwai, Fengtai District, Beijing 100069, China; jia_qi_ruan@hotmail.com (J.R.); liu123zhengyue@163.com (Z.L.); autumn3393@hotmail.com (F.Q.); 17710167797@163.com (H.S.)

**Keywords:** *Artemisia annua* L., ultra-performance liquid chromatography coupled with a photodiode array, sesquiterpene components, quantitative analysis

## Abstract

**Objective**: To develop an accurate and rapid ultra-performance liquid chromatography (UPLC) coupled with a photodiode array (PDA) method for the simultaneous determination of artemisinin (Art), arteannuin B (Art B), arteannuin C (Art C), dihydroartemisinic acid (DHAA) and artemisinic acid (AA) in *Artemisia annua* L. **Methodology**: Chromatography separation was performed on an ACQUITY UPLC BEH C18 Column with isocratic elution; the mobile phase was 0.1% formic acid aqueous solution (A) and acetonitrile (B) (A:B = 40:60, *v*/*v*). Data were recorded at an ultraviolet (UV) wavelength of 191 nm for Art, Art C, DHAA and AA, and 206 nm for Art B. **Results**: The calibration curves of the five sesquiterpene components were all linear with correlation coefficients more than 0.9990. The linear ranges were 31.44–1572 μg/mL, 25.48–1274 μg/mL, 40.56–2028 μg/mL, 31.44–1572 μg/mL and 26.88–1396 μg/mL for Art, Art B, Art C, DHAA and AA, respectively. The precision ranged from 0.08% to 2.88%, the stability was from 0.96% to 1.66%, and the repeatability was all within 2.42% and had a mean extraction recovery of 96.5% to 100.6%. **Conclusion**: The established UPLC–PDA method would be valuable for improving the quantitative analysis of sesquiterpene components in *Artemisia annua* L.

## 1. Introduction

Qinghao, the dry aerial part of the annual Compositae *Artemisia annua* L. (*A. annua*), is widely used as a famous traditional Chinese medicine (TCM). The major biological effects attributed to the constituents of *A. annua* include treating heat, malaria [1] and autoimmune diseases [2]. Previous studies have revealed that there are many kinds of chemical components in Qinghao, and more than 200 components have been found so far, including sesquiterpenoids, flavonoids, coumarins, triterpenoids, and sterols etc [3]. Among them, the most important sesquiterpene components include artemisinin (Art), arteannuin B (Art B), arteannuin C (Art C), dihydroartemisinic acid (DHAA), and artemisinic acid (AA) etc. Art, named Qinghaosu in Chinese, is a sesquiterpene lactone compound showing strong anti-malarial effect, extracted from *A. annua* [4], which contains a natural peroxyl bridge in its chemical structure. It is recognized as one of several natural and effective sesquiterpene compounds with anti-malarial activity [5]. Different harvesting periods, seasons, climates and geographical environments have caused the balance of ingredients in the medicinal materials to be altered. This has affected inherent mutual transformations, and led to observable pharmacological effects. We found that besides Art, the presence of other components and their possible transformation within each other may increase the pharmacological effects of Art, and thus it is necessary to simultaneously detect these five components in order to avoid incomplete quality evaluation.

At present, many analytical methods have been developed for quality control of *A. annua*, including thin layer chromatography (TLC) [6], ultraviolet (UV) spectrophotometry [7], gas chromatography (GC) [8], high performance liquid chromatography (HPLC) [9,10], and so on. Art and Art C are commonly analyzed by HPLC–evaporative light scattering detector (ELSD) [11,12], HPLC–ultraviolet absorption detector [13] and GC–mass spectrometry (GC–MS) [14]; AA and flavonoids are mainly analyzed by HPLC [15,16]. Different methods have their own advantages and disadvantages, for example TLC [6] saves time, has high sensitivity, precision and accuracy of analysis results, but the reproducibility, experimental conditions and data comparability of this method are poor. UV spectrophotometry [7] requires simple instrumentation, but troublesome operation. HPLC–capillary electrophoresis (HPLC–CE) [17] saves the extract for pre-treatment, which is simple, fast and accurate. HPLC [15,16] can analyze and detect terpene lactone, methoxylated flavonoids and AA, but the absorption wavelength and mobile phase of these three types of compounds are different. Although this method is also used for the determination of Art, it also needs to be derivatized, and the experimental steps are cumbersome. HPLC–ELSD [11,12,16] can be used for the determination of Art and Art C, with the components being well-separated despite the complexity of the extract. However, its sensitivity is low and it is not suitable for the analysis of low-content samples. Sometimes the peaks between the compounds interfere with each other. HPLC–MS of the same extract has been shown to unsuccessfully separate Art C and Art, leading to systematic overestimation of Art. GC [8,14] reduces pollution caused by organic solvents, and is environmentally friendly and cost-effective. UV traces obtained on HPLC and liquid chromatography (LC)–MS instruments have been shown to be very similar [18]. Despite obvious changes in retention time, MS is able to detect about one compound, which cannot be detected by 210 nm UV spectroscopy or ELSD. Because of its high sensitivity, analysis of impurities in artemisinin beyond the level set by the current monograph requires the use of MS detection. Ultra-performance liquid chromatography (UPLC) provides good separation in a short period of time, separating the target from possible co-eluting components, and obtaining more accurate results, unlike LC–MS, which is significantly affected by the matrix.

Most of the existing research measures no more than four components in *A. annua*, and no references have been published to simultaneously determine these five sesquiterpenes [19]. Therefore, it is not easy to accurately grasp the transformation and dynamic equilibrium of sesquiterpenes in *A. annua* biosynthesis. The UV or ELSD methods are not sensitive enough, but UPLC uses a smaller particle size column for better resolution and sharper peaks, which improves sensitivity. In general, UV or ELSD methods are inconveniently detected or poorly separated for sesquiterpenoids in *A. annua*, which tends to lead to over-estimation [20,21].

The purpose of our research is to find an accurate and rapid analytical method to simultaneously quantify the contents of Art and its analogues contained in *A. annua.* A dual-wavelength ultra-performance liquid chromatography (UPLC) coupled with a photodiode array (PDA) method has been developed and validated for the quantification of the five sesquiterpene components (Art, Art B, Art C, DHAA and AA) in the extracts of *A. annua* based on their poor solubility in water and high ester solubility. The method shows the characteristics of good separation, high accuracy, and short analysis time. It can provide a reference for objective evaluation of the quality of sesquiterpene components in *A. annua*.

## 2. Results

### 2.1. Single Factor Experiment for Extraction Optimization 

The extraction method, extraction solvent, ratio of material to liquid, and extraction times were taken as the influencing factors, and the total content of Art, Art B, Art C, DHAA and AA was taken as the evaluation indicators. Single factor experiments were carried out to optimize the extraction process. Considering the scoring results and energy cost savings, the optimal extraction process was finally determined to be A1B1C1D1E2. The extraction method was ultrasonic extraction, and used an extraction solvent of methanol over an extraction time of 0.5 h (two cycles of 0.25 h). The ratio of material to liquid was 1:20 (g/mL).

### 2.2. Orthogonal Experiment for Extraction Optimization

The ratio of material to liquid, extraction time and number of extractions were selected as the investigation factors. Referring to the single factor experiment results, the content of five compounds in samples of *A. annua* was used as the index. According to the analysis results, the optimal extraction process conditions were: Extraction with methanol by ultrasonic extraction, comprising of two cycles of 0.25 h each, and a ratio of material to liquid of 1:20 (g /mL).

### 2.3. Optimization of UPLC Separation 

UPLC analysis was performed for separating the five marker compounds in *A. annua*. The mobile phase was a decisive factor in the chromatographic analysis that could affect the separation. To establish an efficient separation of the five compounds, various mobile phases were evaluated, including water, acetonitrile, and methanol with acetic acid. It was found that a 0.1% (*v*/*v*) formic acid aqueous solution (A)–acetonitrile (B) system possessed good separation efficiency with lower noise, and the retention time of the analytes was more stable than the water–methanol system. Higher signal response and sharp peaks of analytes were observed after addition of 0.1% formic acid into the mobile phase, and the sensitivity of detecting was improved. Therefore, 0.1% formic acid–acetonitrile was employed as the final mobile phase with isocratic elution. The UV wavelength used for quantitative analysis was 191 nm for Art, Art C, DHAA and AA, and 206 nm for Art B. Using the established methods of UPLC, the five marker compounds were resolved within 10 min. The retention times of Art, Art B, Art C, DHAA and AA were 3.110, 2.573, 3.256, 6.333 and 7.037 min, respectively. UPLC chromatograms of the *A. annua* extract and the standard compound mixture are shown in Figure 1.

### 2.4. Regression Equation and Linearity

The linear relationships between the concentrations (x, μg/mL) and peak areas (y) of each compound were expressed by the regression equations (y = ax + b) (Table 1). Calibration curves of the marker compounds revealed good linearity with correlation coefficients greater than 0.9990. The results, depicted in Table 1, show that the calibration curves for the five analytes were linear in the range of 31.44–1572 μg/mL for Art, 25.48–1274 μg/mL for Art B, 40.56–2028 μg/mL for Art C, 31.44–1572 μg/mL for DHAA and 26.88–1396 μg/mL for AA.

### 2.5. Precision, Accuracy, Stability, Repeatability and Recovery 

Precision was represented by the relative standard deviations (RSDs) of the concentrations of marker compounds in mixed standard solutions. The results for intra- and inter-day precision and accuracy are shown in Table 2. Intra-day and inter-day precision of the analytes were lower than 2.88%, and the accuracy ranged from 95.1% to 105.3%. The ranges of RSD values for stability of the five compounds in the extract of *A. annua* were from 0.96% to 1.66%. The RSD values for repeatability for the five compounds in the extract of *A. annua* ranged from 0.060% to 0.092% for retention times and from 1.64% to 2.42% for peak areas (Table 3). Recovery tests of the five marker compounds were performed by adding the known amounts of standard solutions to a certain amount of *A. annua* extract. Recovery of each of the five compounds was between 96.5% and 100.6%, with RSD ≤2.82% (Table 4). All values were within acceptable limits, indicating that the established method of UPLC had satisfactory precision, accuracy, repeatability, and recovery for simultaneous analysis.

### 2.6. Determination of Content 

The analytical method developed by UPLC was used to simultaneously quantify five marker compounds from 15 regions of the *A. annua* extract, with three samples in parallel. The experimental results are shown in Table 5. All five compounds were detected in most medicinal materials. The content of Art ranged from 3.649 to 88.79 mg/g; Art B ranged from 0.4597 to 8.593 mg/g; Art C ranged from 0.2121 to 2.967 mg/g; DHAA ranged from 0.4174 to 13.58 mg/g; and AA ranged from 0.4449 to 15.17 mg/g. 

## 3. Discussion

In this study, five sesquiterpene components were simultaneously determined by the UPLC–PDA method. Among the five components, AA was the most abundant compound and Art C was the least abundant compound. The experimental results show that the absorbance of each component—except Art B—was the highest at the wavelength of 191 nm and the absorbance of Art B was the highest at 206 nm. In the total ion current spectra, obtained by TOF MS, the five sesquiterpene components had their most abundant peaks at their protonated molecular ion [M + H]^+^. For Art, Art B, Art C, DHAA and AA, this was found at *m/z* = 283.1534, 249.1490, 267.1619, 237.1847 and 235.1724, respectively, with ppms of −3.9, −0.4, 8.6, −3.4 and 11.1, respectively. 

In the earliest stages of the study, separation effects were investigated using different mobile phase systems, such as methanol–water, acetonitrile–water, acetonitrile–0.1% acetic acid water, 0.1% acetonitrile–0.1% acetic acid water and acetonitrile–0.1% formic acid water. The effects of different organic fractions, specifically from 20% to 80%, on the separation efficiency were optimized under the conditions of isocratic elution and gradient elution, respectively. The results showed that the mobile phase system of acetonitrile–0.1% formic acid water was the best under the condition of isocratic elution. These five sesquiterpene compounds contain acidic compounds, and the addition of an acid to the mobile phase can reduce the degree of dissociation and have good retention properties. Acid addition is a good modifier.

The five compounds were simultaneously determined by an HPLC method in our laboratory before. However, using a common HPLC column, the separation effect is not good, and the separation is better with a smaller particle size of the UPLC column. Therefore, the UPLC method was used to simultaneously quantify these five compounds. In order to verify the accuracy of the UPLC method, the quantitative results of Art and Art B were compared with the quantitative results of a single compound measured by HPLC. The results showed that the UPLC method was 90%–110% of the HPLC results, indicating that the UPLC method was accurate and reliable.

Single factor experiments provide a basis for further accurate orthogonal experiments to extract the materials. Since *A. annua* has no areas with the best yields, this study determined extracts of *A. annua* from 15 regions in China. The preliminary results showed that the contents of five compounds in *A. annua* were quite different, indicating that the contents of sesquiterpenes in *A. annua* in different regions were dissimilar. In the preliminary studies of various components, a peroxide bridge in Art structure was found, which made it the main component in the treatment of malaria. In recent years, more and more researchers—including our team—have paid attention to the therapeutic activities of *A. annua* multi-components in the treatment of malaria [22]. A recent study found that the combination of Art B, AA, scopoletin and Art could improve the efficacy of Art [23,24,25]. Therefore, sesquiterpenoids are the most specific compounds for the treatment of malaria.

In conclusion, the UPLC method established by the current research was used to quantitatively analyze five compounds—Art, Art B, Art C, DHAA and AA—in the extract of *A. annua*. Most of the existing research only measures up to four components in *A. annua*, and no references have been published to simultaneously determine these five sesquiterpenes. Therefore, it is not easy to accurately grasp the transformation and dynamic equilibrium of sesquiterpenes in *A. annua* biosynthesis. The UV or ELSD method is not sensitive enough, but UPLC uses a smaller particle size column for better resolution and sharper peaks, which improves sensitivity [20]. In most of the references, the content is simply measured, but our concern is to provide a basis for the mutual transformation and dynamic balance of the sesquiterpene components in the biosynthesis process through quantitative analysis of *A. annua.* The validation results of the method showed good linearity, repeatability, and intra-day and inter-day precision, stability and recovery, indicating that the method was successfully applied to simultaneous analysis of these five compounds, which provided an analytical means for further quality control of *A. annua*.

## 4. Materials and Methods

### 4.1. Plant Materials

The *A. annua* medicinal materials used in the experiment were all commercial samples purchased from 15 places in China, including Burqin County People’s Hospital, Altay region, Xinjiang Uygur Autonomous Region (N47°50′44.30″E88°08′24.83″) (Xinjiang); Ezhou Hospital of Traditional Chinese Medicine, Hubei Province (N30°23′27.06″ E114°53′41.82″) (Hubei); Quzhou Nankong Chinese Medicine Co., LTD., Zhejiang Province (N28°56′9.31″ E118°52′27.08″) (Zhejiang1); Heze City, Shandong Province (N35°14′1.07″ E115°28′52.14″) (Shandong1); Shengzhou City, Zhejiang Province (N29°35′18.56″ E120°49′18.12″) (Zhejiang2); Anguo Baicao Medicinal Material Shop, Hebei Province (N38°25′6.42″ E115°19′35.90″) (Hebei); Yan’an City, Shaanxi Province (N36°35′7.04″ E109°29′23.21″) (Shaanxi); Meitan County, Zunyi City, Guizhou Province (N27°44′56.00″ E107°27′55.84″) (Guizhou); Xinning County, Shaoyang City, Hunan Province (N26°26′0.46″ E110°51′24.26″) (Hunan1); Capital Medical University Chinese Medicine Clinic-Beijing Herborist Chinese Herbal Pieces Co., Ltd., Beijing City (N39°54′16.88″ E116°24′25.81″) (Beijing1); Jinan Hebao Chinese Medicine Co., Ltd., Shandong Province (N36°33′14.50″ E116°45′7.16″) (Shandong2); Self-collection, Beijing City (N39°54′16.88″ E116°24′25.81″) (Beijing2); Longshan County Medicinal Materials Market in Xiangxi Autonomous Prefecture, Hunan Province (N28°18′44.00″ E109°44′20.15″) (Hunan2); Changtu County, Tieling City, Liaoning Province (N42°47′10.07″ E124°06′39.31″) (Liaoning); Medicinal materials provided by the pharmaceutical company, Hunan Province (N26°25′13.22″ E111°36′44.10″) (Hunan3). They were all identified by Professor Manyuan Wang (School of Traditional Chinese Medicine, Capital Medical University) as the dry ground part of *Artemisia annua* L. and stored at room temperature and protected from light.

### 4.2. Chemicals and Reagents

The reference standards of Art, Art B, Art C, DHAA and AA (chemical structures shown in Figure 2) were all isolated from *A. annua* in our laboratory (School of Traditional Chinese Medicine, Capital Medical University) [10,22]. Their chemical structures were further confirmed by UV, IR, ^1^H NMR, ^13^C NMR and HR–ESI–MS techniques. Their purities were also evaluated to be all greater than 99.0% by the HPLC–DAD method at UV 210 nm using the area normalization procedure. The conditions for mass spectrometry detection were as follows. The mass spectrometric analysis was performed on an LC–ESI–TOF equipped with an ESI source (#SYNAPT G2-Si, Waters Corporation, Milford, MA, USA). All acquisitions were performed under positive ionization mode with a capillary voltage of +1000 V. The source temperature was 120 °C, and nitrogen was used as nebulizer and desolvation gas, as well as the drying gas at 800 L·h^−1^ at a desolvation temperature of 450 °C. The cone gas was 50 L·h^−1^; the sampling cone was 40 V and the collision energy (CE) was 4 V. 8 V and 12 V was selected as the trap collision energy. The acquisition mass range was from *m/z* 50 to *m/z* 1000. The total run time for analysis was 10 min (TOF MS total ion current spectra shown in Figure 3). HPLC-grade acetonitrile was purchased from Fisher Chemical Co. (Reagent Lawn, Nanjing, Jiangsu Province, China); purified water was purchased from Hangzhou Wahaha Group Co. (Hangzhou, Zhejiang Province, China); analytical-grade formic acid, methanol, petroleum ether and ethyl acetate were all purchased from Beijing Chemical Works (Beijing, China); and sodium hydroxide was purchased from Modern Oriental (Beijing) Technology Development Co. (Beijing, China).

### 4.3. Apparatus and Chromatographic Conditions

An Agilent 1290 UPLC system (Agilent Technologies, Santa Clara, CA, USA) equipped with a pump, degasser, column oven, auto sample injector, and PDA detector (#1290, Agilent Technologies, Santa Clara, CA, USA) was used for liquid chromatography analysis. The chromatographic separation of the five marker compounds was performed at 45°C using an ACQUITY UPLC BEH C18 Column (2.1 mm × 100 mm, 1.7 μm, Waters Corp., Milford, MA, USA) and ACQUITY UPLC C18 Van Guard Pre-column (2.1 mm × 5 mm, 1.7 μm, Waters Corp., Milford, MA, USA) with an isocratic elution at a flow rate of 0.20 mL/min, using a mobile phase consisting of 0.1% formic acid (A) and acetonitrile (B) (A:B = 40:60, *v*/*v*).The injection volume was 1 μL. Data were recorded at a UV wavelength of 191 nm for Art, Art C, DHAA and AA, and 206 nm for Art B.

### 4.4. Preparation of Standard Solutions

In a clean, dry 5 ml volumetric flask, each reference standard (approximately 5 mg) of Art, Art B, Art C, DHAA and AA was accurately weighed and dissolved in HPLC grade acetonitrile to make a stock solution. Calibration standards were prepared by diluting the stock solution with acetonitrile in appropriate quantities.

### 4.5. Preparation of the Extracts and Sample Solutions

The extraction method (A), extraction solvent (B), ratio of material to liquid (C), extraction time (D), and cycle of extraction (E) were taken as the influencing factors, while the respective content and total content of Art, Art B, Art C, DHAA and AA were taken as the comprehensive evaluation indicators. Single factor experiments were carried out to optimize the extraction process. The level of the extraction process factors was shown in Table 6.

The ratio of material to liquid, extraction time and number of extractions were selected as the investigation factors. Referring to the single factor experiment results, two types of content (of the five compounds in the samples of *A. annua*) were used as the index, and the L (3^3^) orthogonal table design test plan was used to determine the optimum extraction conditions of the *A. annua*. The optimum process conditions for extracting are shown in Table 7. The design scheme of the orthogonal experiment is shown in Table 8.

*A. annua* was accurately weighed to 1.0 g and 20 volumes of methanol were added to the sample. After 0.25 h ultrasonic extraction time, the extract was filtered through a quantitative filter paper (12.5 cm). The filtrate was concentrated in vacuum by a rotary evaporator (EYELA N-1001, Rikakikai Co., Tokyo, Japan) system to evaporate the sample and prepare a paste extract (0.1405 g). The yield of *Artemisia annua* L. ether extract was 14.1%. The extract was accurately weighed, and the volume of acetonitrile was adjusted to 5 mL to obtain a final extract of *A. annua*. The sample solution was then filtered through a syringe filter (0.22 μm) and used for UPLC analysis.

### 4.6. Calibration Curve and Lowest Limit of Quantification (LLOQ)

The five reference standard solutions were diluted 2, 5, 10, 20, and 50 times with acetonitrile, filtered through a 0.22 μm microporous membrane, and injected sequentially. The calibration curves were drawn with the sample concentration (μg/mL) as the abscissa and the peak area as the longitudinal coordinate. The concentration ranges of the reference substances were as follows: Art (1572–31.44 μg/mL), Art B (25.48–1274 μg/mL), Art C (40.56–2028 μg/mL), DHAA (31.44–1572 μg/mL) and AA (26.88–1396 μg/mL). LLOQ was defined as the lowest concentration on the calibration curve with an acceptable accuracy within ± 5% and precision less than 5% [26].

### 4.7. Precision, Accuracy, Stability, Repeatability and Recovery

The precision of the UPLC method was evaluated by using a mixed standard solution of marker compounds at low, medium and high levels to measure intra-day and inter-day variations [27]. The mixed standard solution was analyzed five times in a single day to determine intra-day precision, and the inter-day precision of the method was evaluated by repeating the injection of the standard solution for three consecutive days. The method was as follows: 1 μL of the mixed reference solution was drawn and continuously injected, and the peak area of the chromatogram recorded. The intra- and inter-day RSD (%) was used to indicate precision, while the percentage obtained from dividing the observed concentration by the fortified concentration x100% was used to indicate accuracy. Specifically, the same sample solution was taken and injected at 1, 2, 4, 8, 12, and 24 h. The RSD (%) of the peak area was used to indicate the stability of the sample solution. The RSD of the peak areas of the five marker components were all less than 2%, indicating that the sample solution was stable within 24 h. Five replicates of the same *A. annua* sample were prepared to act as the sample solutions, according to the method described in Section 4.5, and the injection phase was repeated six times to evaluate the repeatability of RSD (%) for retention time and peak area. The results showed that RSD was less than 3%, indicating that the method was reproducible. Additionally, six samples of *A. annua* with known content were accurately weighed and a certain concentration of standard solution added into the sample. This was prepared in accordance with the method of Section 4.5. These samples were then injected to determine the recovery of the five compounds. The recovery rate was repeated 6 times.

## Figures and Tables

**Figure 1 molecules-24-01530-f001:**
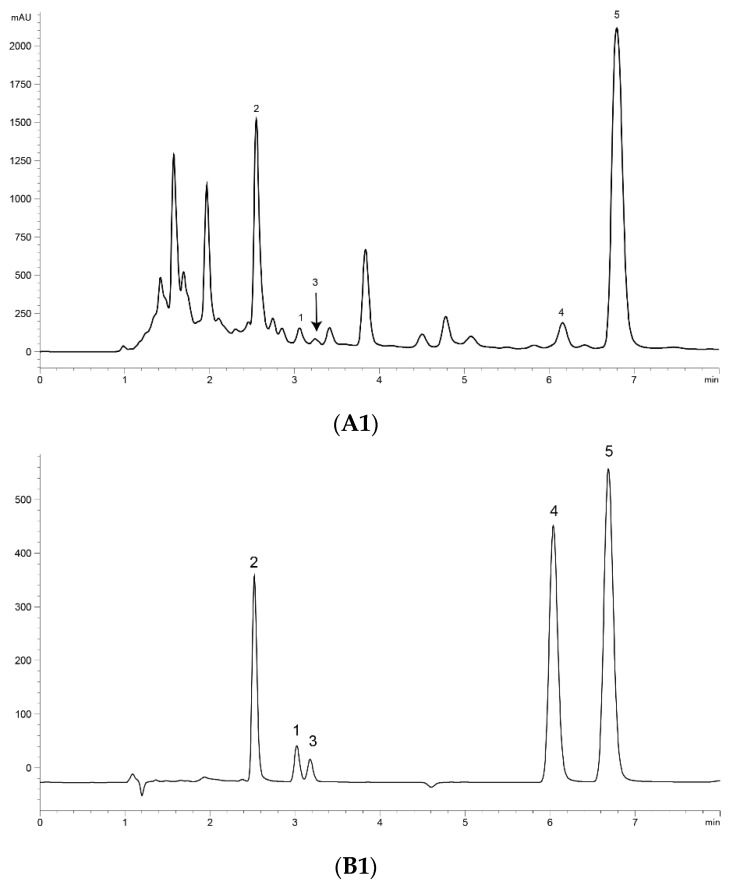
Ultra-performance liquid chromatography (UPLC) chromatograms of the extract of *Artemisia annua* L. (*A. annua*) (A) and a standard mixture (B). (**A1**) (**B1**) at 191 nm and (**A2**) (**B2**) at 206 nm. (1) Artemisinin (Art); (2) arteannuin B (Art B); (3) arteannuin C (Art C); (4) dihydroartemisinic acid (DHAA); and (5) artemisinic acid (AA).

**Figure 2 molecules-24-01530-f002:**
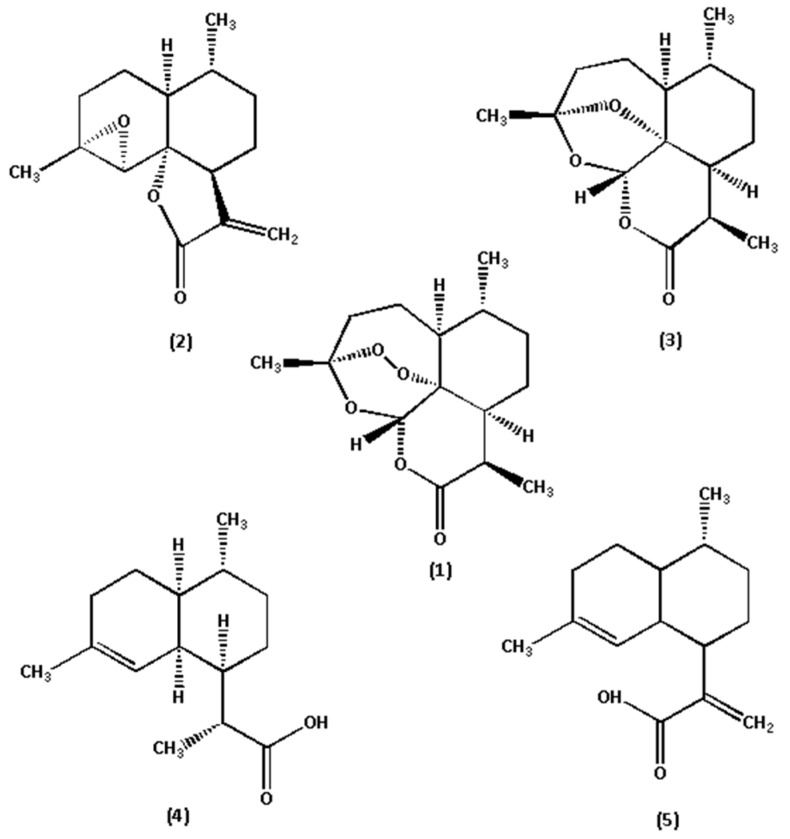
Chemical structures of the five marker components in *A. annua*. (1) Art, (2) Art B, (3) Art C, (4) DHAA, and (5) AA.

**Figure 3 molecules-24-01530-f003:**
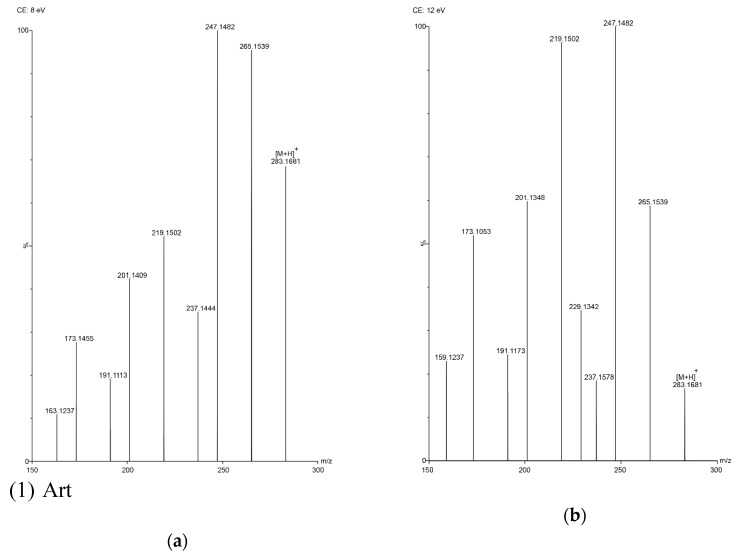
TOF MS total ion current spectra of the five compounds in two fragmentation energies. (1) Art, *m/z* = 283.1534; (2) Art B, *m/z* = 249.1490; (3) Art C, *m/z* = 267.1619; (4) DHAA, *m/z* = 237.1847; (5) AA, *m/z* = 235.1724. a: The lower fragmentation energy (8 V); b: The higher fragmentation energy (12 V).

**Table 1 molecules-24-01530-t001:** Regression equations, correlation coefficients and linear ranges for compounds.

Compound	Linear Range (μg/mL)	Regression Equation (y = ax + b) ^a)^		R	LLOQ ^b)^ (μg/mL)
		Slope (a)	Intercept (b)		
Art	31.44–1572	0.8332	7.996	1.000	31.44
Art B	25.48–1274	8.265	109.9	0.9996	25.48
Art C	40.56–2028	0.4145	1.707	0.9998	40.56
DHAA	31.44–1572	9.186	38.13	0.9998	31.44
AA	26.88–1396	14.28	175.6	0.9990	26.88

^a)^ y = ax + b, y means peak area and x means concentration (µg/mL). ^b)^ LLOQ means the lowest limit of quantification.

**Table 2 molecules-24-01530-t002:** Precision and accuracy of compounds in *A. annua.*

Compound	Fortified Conc. (μg/mL)	Intra-Day (n = 5)			Inter-Day (n = 5)		
		Observed Conc. (μg/mL)	Precision ^a)^ (%)	Accuracy ^b)^ (%)	Observed Conc. (μg/mL)	Precision (%)	Accuracy (%)
Art	786.0	817.6	0.21	104.0	807.9	1.27	102.8
	314.4	330.1	0.15	105.0	324.3	2.42	103.1
	157.2	163.6	0.09	104.1	162.4	1.89	103.3
Art B	637.0	660.3	0.13	103.7	605.8	0.32	95.10
	254.8	266.5	0.22	104.6	246.9	0.24	96.90
	127.4	129.8	0.10	101.9	123.0	1.12	96.60
Art C	1014	1068	0.63	105.3	1037	2.16	102.2
	405.6	391.6	1.49	96.60	395.6	2.66	97.50
	202.8	212.9	0.18	105.0	197.4	2.88	97.30
DHAA	786.0	821.4	0.08	104.5	821.4	1.69	104.5
	314.4	327.3	0.17	104.1	321.9	2.34	102.4
	157.2	161.7	0.15	102.9	158.3	2.85	100.7
AA	672.0	701.9	0.15	104.4	705.6	2.76	105.0
	268.8	281.7	0.19	104.8	280.8	2.10	104.5
	134.4	136.0	0.24	101.2	130.3	2.37	97.00

^a)^ Precision is expressed as relative standard deviation (RSD) (%) = (SD/Mean) × 100; ^b)^ Accuracy (%) = (Observed concentration/Fortified concentration) × 100.

**Table 3 molecules-24-01530-t003:** Stability and repeatability for compounds in *A. annua.*

Compound	Stability	Repeatability	RSD (%)
	RSD (%)	Retention Time (min)	Peak Area (AU)
Art	1.63	0.069	2.37
Art B	1.65	0.060	2.42
Art C	1.52	0.070	2.04
DHAA	1.45	0.091	2.15
AA	0.96	0.092	1.64

**Table 4 molecules-24-01530-t004:** Recovery of compounds in *A. annua* (n = 6).

Compound	Original Conc.	Spiked Conc.	Found Conc.	Recovery ^a)^ ± SD	RSD (%)
	(μg/mL)	(μg/mL)	(μg/mL)	(%)	
Art	116.1	117.0	233.7	100.6 ± 1.28	1.27
Art B	179.5	180.0	358.8	99.6 ± 0.83	0.84
Art C	10.70	13.00	23.23	96.5 ± 2.72	2.82
DHAA	32.90	33.00	65.70	99.4 ± 2.67	1.69
AA	282.5	285.0	561.6	97.9 ± 2.50	2.56

^a)^ Recovery (%) = (Found concentration – Original concentration)/spiked concentration × 100.

**Table 5 molecules-24-01530-t005:** The content of compounds in the *A. annua* extract from 15 *A. annua* ($\bar{x}$ ± SD).

No.	Region	Art Content (mg/g)	Art B Content (mg/g)	Art C Content (mg/g)	DHAA Content (mg/g)	AA Content (mg/g)
1	Xinjiang	10.16 ± 0.011	5.347 ± 0.0023	2.700 ± 0.0025	-	3.449 ± 0.0015
2	Hubei	12.04 ± 0.017	8.108 ± 0.0015	2.228 ± 0.0021	0.8521 ± 0.00025	15.17 ± 0.0072
3	Zhejiang1	5.457 ± 0.0031	4.581 ± 0.0020	1.153 ± 0.0016	-	6.459 ± 0.0016
4	Shandong1	8.604 ± 0.0022	8.356 ± 0.0025	0.9970 ± 0.00036	0.6356 ± 0.00030	7.735 ± 0.0021
5	Zhejiang2	27.59 ± 0.015	4.462 ± 0.0015	0.9026 ± 0.00021	-	3.022 ± 0.0019
6	Hebei	88.79 ± 0.010	1.289 ± 0.0020	-	-	1.406 ± 0.0021
7	Shaanxi	7.594 ± 0.0021	8.593 ± 0.0011	1.224 ± 0.0025	-	7.620 ± 0.0021
8	Guizhou	13.78 ± 0.010	3.332 ± 0.0015	1.980 ± 0.0019	-	4.499 ± 0.0020
9	Hunan1	8.216 ± 0.0012	0.4597 ± 0.00070	0.9990 ± 0.00022	2.020 ± 0.0025	0.4449 ± 0.00024
10	Beijing1	5.562 ± 0.0010	6.877 ± 0.0020	1.379 ± 0.0022	3.737 ± 0.0022	8.313 ± 0.0013
11	Shandong2	4.457 ± 0.0020	7.391 ± 0.0014	1.494 ± 0.0016	0.4174 ± 0.00020	11.14 ± 0.0034
12	Beijing2	2.303 ± 0.0019	3.562 ± 0.0021	0.2121 ± 0.00010	0.6527 ± 0.0019	5.606 ± 0.0024
13	Hunan2	6.880 ± 0.0026	0.5579 ± 0.00041	0.8605 ± 0.00045	3.587 ± 0.0019	0.5311 ± 0.0078
14	Liaoning	3.649 ± 0.0016	6.323 ± 0.0022	1.338 ± 0.0017	-	6.477 ± 0.0029
15	Hunan3	63.74 ± 0.010	4.028 ± 0.0015	2.967 ± 0.0015	13.58 ± 0.0057	3.231 ± 0.0021

**Table 6 molecules-24-01530-t006:** The level of extraction process factors.

Level	Factors				
	A Method	B Solvent	C Ratio (g/mL)	D Time/h	E Cycles
1	ultrasonic	methanol	1:20	0.5	1
2	Condensation reflux	petroleum ether	1:30	1	2
3	-	ethyl acetate	1:40	2	3

**Table 7 molecules-24-01530-t007:** The level of orthogonal experimental factors.

Level	Factors		
	X ratio/(g/mL)	Y time/h	Z times
a	1:10	0.25	1
b	1:20	0.5	2
c	1:30	1	3

**Table 8 molecules-24-01530-t008:** The design scheme of orthogonal experiment.

No.	X	Y	Z
1	a	a	a
2	a	b	b
3	a	c	c
4	b	a	b
5	b	b	c
6	b	c	a
7	c	a	c
8	c	b	a
9	c	c	b

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
