# Peer review of "Simultaneous Quantification of Five Sesquiterpene Components after Ultrasound Extraction in Artemisia annua L. by an Accurate and Rapid UPLC–PDA Assay†"

_molecules, 2019, doi:10.3390/molecules24081530_

Round 1
Reviewer 1 Report
It is opinion of the reviewer that this paper before acceptance Leeds several corrections. My individual comments are listed below.
Information about the method of extraction must be added to a paper title.
References must be cited as “[...]”.
L. 20 – It should be “..and the contents …”.
L. 39 – Remove italic for “Qinghao”.
L. 52 – It must be “chromatography” instead of “scanning”.
L. 55 – It should be “GC-mass spectrometry”.
L. 59 – Full name of “EC”.
L. 65 – It should be “analysis” instead of “determination”.
L. 66 – How peaks can interfere?
L. 73 – What kind of the physical and chemical properties?
L. 79 – Plant name must be in italic.
L. 99 – What does it mean “self-made”? Synthesis of the five markers must be described.
Information about condition of TOF MS must be added to the Methods.
I suggest describe extraction before UHPLC determination.
L. 154 – It should be “cycle of extraction (E)”.
L. 274 – The content cannot be exhibited as %!
Table 7 – Results of ±SD must be added.
The results should be described using PCA.
L. 329 - Plant name must be in italic.
L. 363 – It should be “J. Agric. Food Chem.”.
L. 382 – It should be “Molecules”.
Author Response
1. Information about the method of extraction must be added to a paper title.
Response: The title has been corrected to be “Simultaneous Quantification of Five Sesquiterpene Components after Ultrasound Extraction in Artemisia annua L. by an Accurate and Rapid UPLC-PDA Assay”.
2. References must be cited as “[...]”.
Response: We have made all the corrections according to your comment.
3. L. 20 – It should be “…and the contents …”.
Response: Thank you for your kindly suggestion, and this description has been deleted because of the simplification of our Abstract.
4. L. 39 – Remove italic for “Qinghao”.
Response: The italic writing for “Qinghao” has been changed.
5. L. 52 – It must be “chromatography” instead of “scanning”.
Response: This description has been corrected.
6. L. 55 – It should be “GC-mass spectrometry”.
Response: This description has been revised.
7. L. 59 – Full name of “CE”.
Response: The full name of “CE” has been added.
8. L. 65 – It should be “analysis” instead of “determination”.
Response:This description has been corrected.
9. L. 66 – How peaks can interfere?
Response: We have explained how peaks can interfere in the section “1 Introduction”.
10. L. 73 – What kind of the physical and chemical properties?
Response: The physical and chemical properties of the components have been added in the section “1 Introduction”.
11. L. 79 – Plant name must be in italic.
Response: The plant name has been changed into italic.
12. L. 99 – What does it mean “self-made”? Synthesis of the five markers must be described.
Response: We have re-written this description in the section “4.2 Materials and Methods” according to your suggestion.
13. Information about condition of TOF MS must be added to the Methods.
Response: The information about the conditions of TOF MS has been added to the section “4.2 Materials and Methods” .
14. I suggest describe extraction before UHPLC determination.
Response: The section “2.5 Preparation of the Extracts and Sample Solutions” has been revised to be “4.5” in Materials and Methods. The corresponding revisions have been made on those sections “4.6. Calibration Curve and LLOQ” “4.7. Precision, accuracy, stability, repeatability and recovery” in Materials and Methods and “2.1. Single Factor Experiment for the extraction optimization” “2.2. Orthogonal experiment for the extraction optimization” “2.3. Optimization of UPLC Separation” “2.4. Regression Equation and Linearity” in Results.
15. L. 154 – It should be “cycle of extraction (E)”.
Response: This description has been corrected.
16. L. 274 – The content cannot be exhibited as %!
Response: All the contents have been revised as mg/g (average±SD).
17. Table 7 – Results of ±SD must be added.
Response: The results of ±SD have been added in those corresponding tables.
18. The results should be described using PCA.
Response: Thank you for your kindly suggestion. In the course of the experiment, we tried to analyze the experimental data by principal component analysis (PCA). However, the statistical results showed that only a part of the sample data meets the requirements of reasonable structure and cannot achieve good practicability. Therefore, we abandoned this method for describing the results.
19. L. 329 - Plant name must be in italic.
Response: The plant name has been described in italic.
20. L. 363 – It should be “J. Agric. Food Chem.”.
Response: This Journal name has been corrected.
21. L. 382 – It should be “Molecules”.
Response: This Journal name has been corrected.
Reviewer 2 Report
Dear Authors, having read your nicely described and well prepared manuscript i have the following comments:
- in your manuscript I miss the relation of your work to what is already done. i would recommend to develop the introduction section and after the described analytical techniques used for the evaluation of Artemisia metabolites, please, add some more details on the analyses performed by other researchers using LC-MS facilities.
- underline the novelty of your data and their impact on the analysis of Artemisia species
- the discussion should be also written in relation to other studies - what did the authors achieve that is better than other authors.
- create a supplementary file and move there figure 2, which delivers data on the purity of single components
- i would recommend you to correct the figure 3. with TOF-MS in the TIC sectra presented in the manuscript we see many peaks - we do not know actually where is the molecular ion. Please, show the TIC chromatograms in two fragmentation energies - a llower, which will show the molecular ion and in a higher to visualize the fragments. please mark the signals which are the characteristic for your compounds. Like this everyone will have impression, that these spectra come from dirty compounds.
- also, increase the resolution of the graphs - make the numbers bigger, easier to read
Other minor comments:
- please, rewrite the line 164 - 'ultrasonic for 0.25h' is incorrect
- line 179: give reference to the method of LLOQ calculation
- section 2.7. - rewrite the section in a non-personal manner, e.g. line 192: 'take the same batch' is incorrect - here we have to be more objective and relate to what was done and no to what we have to do
Author Response
1. In your manuscript I miss the relation of your work to what is already done. I would recommend to develop the introduction section and after the described analytical techniques used for the evaluation of Artemisia metabolites, please, add some more details on the analyses performed by other researchers using LC-MS facilities.
Response: We have made some corrections in the section “1 Introduction” according to your comments and 2 references have also been added in our manuscript in order to further clarify the relation of our work to these published research results.
2. Underline the novelty of your data and their impact on the analysis of Artemisia species.
Response: The novelty of the data and their impact on the analysis has been described in the section “3 Discussion”.
3. The discussion should be also written in relation to other studies - what did the authors achieve that is better than other authors.
Response: According to your suggestion, we have re-written the section “3 Discussion” in relation to other studies.
4. Create a supplementary file and move there figure 2, which delivers data on the purity of single components.
Response: Figure 2 has been deleted in our manuscript and moved it into a separate supplementary file.
5. I would recommend you to correct the figure 3 with TOF-MS in the TIC spectra presented in the manuscript we see many peaks - we do not know actually where is the molecular ion. Please, show the TIC chromatograms in two fragmentation energies - a lower, which will show the molecular ion and in a higher to visualize the fragments. Please mark the signals which are the characteristic for your compounds. Like this everyone will have impression, that these spectra come from dirty compounds.
Response: We have already added the TIC chromatograms in two fragmentation energies - a lower (8 V), which will show the molecular ion and in a higher (12 V) to visualize the fragments and mark the signals ([M+H]+) which are the characteristic for the five marker components. The signals for the marker compounds was added in the TIC chromatograms.
6. Increase the resolution of the graphs - make the numbers bigger, easier to read.
Response: The resolutions of all the graphs have been made bigger and easier to read.
7. Rewrite the line 164 - 'ultrasonic for 0.25h' is incorrect.
Response: This description has been corrected.
8. Line 179: give reference to the method of LLOQ calculation.
Response: One reference to the method has been added.
9. Section 2.7. - rewrite the section in a non-personal manner, e.g. line 192: 'take the same batch' is incorrect - here we have to be more objective and relate to what was done and no to what we have to do.
Response: We have re-written this section “4.7 Methoods” according to your suggestion.
Reviewer 3 Report
22- March -2019
Journal: Molecules
Manuscript ID: molecules-477771-peer-review
Title: Simultaneous Quantification of Five Sesquiterpene Components in Artemisia annua L. by an Accurate and Rapid UPLC-PDA Assay
Authors: Jiaqi Ruan, Zhengyue Liu, Feng Qiu, Henan Shi and Manyuan Wang
Dear Editor:
The authors have investigated the simultaneous quantification of five sesquiterpene components in artemisia annua L. using an accurate and rapid UPLC-PDA assay. This is a good study indicating the precision, accuracy, stability, repeatability and recovery of UPLC-PDA assay. The manuscript is adequate for publication after some minor changes which are indicated below.
Hesham El-Seedi, professor,
Division of Pharmacognosy
Department of Medicinal Chemistry
Uppsala University, Biomedical Centre
Box 574, SE-75 123, Uppsala, Sweden
Tel: +46-18-4714207
Fax: +46-18-509101
E-mail: hesham.el-seedi@ilk.uu.se
Comments to Authors
Abstract:
1- The authors should add high quality graphical abstract.
2- The abstract shouldn`t be more than 200 words.
3- The abstract describes some steps of the experiment, so the authors would rewrite the abstract in a briefly form.
4- The originality and the aim are missing.
Keywords:
The authors wouldn`t add technique abbreviation in keywords list as (UPLC-PDA).
Introduction:
1- The authors would add the importance of natural products especially from plants.
2- Please add the traditional uses of this plant (Artemisia annua L) in TCM.
3- What about the biological activities of sesquiterpenes.
4- The authors could add the advantages of UPLC-PDA over other traditional techniques such as (HPLC-CE and HPLC-UV).
5- “so it is necessary to simultaneously detect these five components simultaneously in order to avoid incomplete quality evaluation” please explain in more details
6- What do you mean by “Art and Art C”
7- The introduction should be revised thoroughly
8- The authors could benefit from the following reference in the introduction:
Ali, S.E., El Gedaily, R.A., Mocan, A., Farag, M.A. and El-Seedi, H.R., 2019. Profiling Metabolites and Biological Activities of Sugarcane (Saccharum officinarum Linn.) Juice and its Product Molasses via a Multiplex Metabolomics Approach. Molecules, 24(5), p.934.
Methodology:
1- The authors should put the results and discussion part before material and methods in accordance to the journal instruction.
2- Material and Methods seem to be suitable for the objectives of this study.
3- In figure 2: the authors would clarify the name of the technique used for measuring purity.
4- In figure 3: the numbers that appear on this figure are not clear.
5- In preparation of the extracts and sample solutions: the authors would add indication of L (33).
6- From abstract and introduction: the authors evaluated the (UPLC) coupled with photodiode array (PDA) but in methods didn`t explain this coupling between UPLC and PDA.
Results:
1- The authors would explain (It was found that 0.1% (v/v) formic acid aqueous solution (A)-acetonitrile (B) system possessed good separation efficiency)
2- The authors would explain this point (single factor experiment for the extraction optimization).
3- The authors would identify (intra-day and inter-day precisions).
Discussion:
1- The authors didn`t indicate through this experiment how the combination of Art B, AA, and scopoletin could improve the efficacy of Art.
2- The authors didn`t use HPLC for indicating the compounds in this experiment, so the authors would add references for lines from 296 to 300.
References:
1- The authors would check the style of writing references in accordance to journal instruction.
Author Response
Abstract:
1. The authors should add high quality graphical abstract.
Response: The graphical abstract has been revised and moved it into a separate supplementary file.
2. The abstract shouldn`t be more than 200 words.
Response: The abstract has been re-written to less than 200 words (193 words).
3. The abstract describes some steps of the experiment, so the authors would rewrite the abstract in a briefly form.
Response: The abstract has been re-written according to your suggestion.
4. The originality and the aim are missing.
Response: The section “1 Introduction” has been re-written, hoping to highlight the innovation of the article, and the purpose of this article is more clear.
Keywords:
5. The authors wouldn`t add technique abbreviation in keywords list as (UPLC-PDA).
Response: This description has been corrected.
Introduction:
6. The authors would add the importance of natural products especially from plants.
Response: The importance of natural products has been added in this section “1 Introduction”.
7. Please add the traditional uses of this plant (Artemisia annua L) in TCM.
Response: The traditional uses of the plant has been added in this section “1 Introduction”.
8. What about the biological activities of sesquiterpenes.
Response: The biological activities of sesquiterpenes have been added.
9. The authors could add the advantages of UPLC-PDA over other traditional techniques such as (HPLC-CE and HPLC-UV).
Response: The advantages of UPLC have been added in the section “1 Introduction”.
10. Please explain in more details “so it is necessary to simultaneously detect these five components simultaneously in order to avoid incomplete quality evaluation”.
Response: This part in section “1 Introduction” has been re-written by adding more descriptions.
11. What do you mean by “Art and Art C” .
Response: L. 35 and L. 37 has explained what they are as follows: “Among them, most important sesquiterpene components include artemisinin (Art), arteannuin B (Art B), arteannuin C (Art C), dihydroartemisinic acid (DHAA), artemisinic acid (AA) etc.”.
12. The introduction should be revised thoroughly.
Response: According to your suggestion, the introduction section has been revised thoroughly.
13. The authors could benefit from the following reference in the introduction:
Ali, S.E., El Gedaily, R.A., Mocan, A., Farag, M.A. and El-Seedi, H.R., 2019. Profiling Metabolites and Biological Activities of Sugarcane (Saccharum officinarum Linn.) Juice and its Product Molasses via a Multiplex Metabolomics Approach. Molecules, 24(5), p.934.
Response: Thanks for providing the reference to make us benefit from it. The introduction section has been re-written according to the reference.
Methodology:
14. The authors should put the results and discussion part before material and methods in accordance to the journal instruction.
Response: The section “Results” and “Discussion” has been put before the section “Materials and Methods” in accordance to the journal instruction. The numbers of tables, figures and references have been all changed.
15. Material and Methods seem to be suitable for the objectives of this study.
Response: The description of the title has been changed.
16. In figure 2: the authors would clarify the name of the technique used for measuring purity.
Response: The name of the technique used for measuring purity has been added and the peak purity chromatograms of these five compounds by HPLC-DAD method has been added in figure 2. All of the figures have been moved into an supplementary file.
17. In figure 3: the numbers that appear on this figure are not clear.
Response: The resolutions of all the figures have been made bigger and easier to read.
18. In preparation of the extracts and sample solutions: the authors would add indication of L (33).
Response: The design scheme of orthogonal experiment has been added.
19. From abstract and introduction: the authors evaluated the (UPLC) coupled with photodiode array (PDA) but in methods didn`t explain this coupling between UPLC and PDA.
Response: DAD and PDA means the same detector, This description “DAD” has been corrected as “PDA”.
Results:
20. The authors would explain (It was found that 0.1% (v/v) formic acid aqueous solution (A)-acetonitrile (B) system possessed good separation efficiency).
Response: The explanation has been added in the section “3. Discussion”.
21. The authors would explain this point (single factor experiment for the extraction optimization).
Response: The explanation has been added in the section “3. Discussion”.
22. The authors would identify (intra-day and inter-day precisions).
Response: The reference of intra-day and inter-day preisions was added.
Discussion:
23. The authors didn`t indicate through this experiment how the combination of Art B, AA, and scopoletin could improve the efficacy of Art.
Response: The experiment of how the combination of Art B, AA, and scopoletin could improve the efficacy of Art has ever been studied in our laboratory. Here we are going to provide published articles about this experiment to demonstrate the accuracy of what we describe.
Zhang, C.; Gong, M. X.; Qiu, F.; Li, J.; Wang, M. Y. Effects of arteannuin B, arteannuic acid and scopoletin on pharmacokinetics of artemisinin in mice. Asian Pac. J. Trop. Med. 2016, 9, 677–681.
24. The authors didn`t use HPLC for indicating the compounds in this experiment, so the authors would add references for lines from 296 to 300.
Response: The explanation has been added in the section “3. Discussion”.
25. The authors would check the style of writing references in accordance to journal instruction.
Response: The style of writing references in accordance to journal instruction has been checked.
Round 2
Reviewer 1 Report
The authors corrected this paper properly taken under considerations all my comments. Therefore I can accept it now.
Reviewer 2 Report
Dear Authors
Thank you for your corrections. In the figure 2 you can easily align the spectra as two in one line
I have no more comments to your text